# Room-Temperature Infrared Photodetectors with Zero-Dimensional and New Two-Dimensional Materials

**Taipeng Li** [1], **Xin Tang** [1,2,3] **and Menglu Chen** [1,2,3,*]

1   School of Optics and Photonics, Beijing Institute of Technology, Beijing 100081, China;
    3120210626@bit.edu.cn (T.L.); xintang@bit.edu.cn (X.T.)
2   Beijing Key Laboratory for Precision Optoelectronic Measurement Instrument and Technology,
    Beijing 100081, China
3   Yangtze Delta Region Academy of Beijing Institute of Technology, Jiaxing 314019, China
*   Correspondence: menglu@bit.edu.cn

**Abstract:** Infrared photodetectors have received much attention for several decades due to their broad applications in the military, science, and daily life. However, for achieving an ideal signal-to-noise ratio and a very fast response, cooling is necessary in those devices, which makes them bulky and costly. Thus, room-temperature infrared photodetectors have emerged as a hot research direction. Novel low-dimensional materials with their easy fabrication and excellent photoelectronic properties provide a possible solution for room-temperature infrared photodetectors. This review aims to summarize the preparation methods and characterization of several low-dimensional materials (PbS, PbSe and HgTe, new two-dimensional materials) with great concern and the room-temperature infrared photodetectors based on them.

**Keywords:** infrared photodetectors; room temperature; low-dimensional





## 1. Introduction

In nature, every object above absolute zero can emit infrared photons with unique fingerprints of different temperature information, though they cannot be captured by human eyes. Infrared detectors, however, can convert infrared radiation into a measurable electronic signal, which is significant in infrared technology [1]. The technical progress of infrared technology is mainly related to the development of narrow-band semiconductor infrared photodetectors, which have both an ideal signal-to-noise ratio and a very fast response. However, to achieve this goal, the photodetectors need to be cooled at low temperatures to avoid thermal-activated carriers. As a result, the system is bulky, costly and inconvenient to use. As a result, developing room-temperature infrared photodetectors has become an important research direction [1].

Driven by Moore's Law, the feature size of highly integrated electronics has already been reduced to a few nanometers [2]. Consequently, in infrared technology, one major focus is how to reduce pixel scale and increase format array for a highly integrated photoelectric detection system with high performance. Traditional infrared photodetectors based on narrow-band semiconductors, such as indium–gallium arsenide (InGaAs) and mercury–cadmium telluride (HgCdTe), need the flip-up bonding technique. This causes great difficulties in coupling with silicon electronics, limiting the development of the focal plane array (FPA). Low-dimensional nanostructured materials are appealing in the field of photoelectric detection because they can be integrated with traditional silicon electronics and even with flexible large-area substrates by liquid phase processing, such as spin-coating, spraying, or stratified deposition [3]. In addition, they have great potential in subwavelength pixel, large array and multicolor devices [4]. However, there are still many challenges in material growth, device fabrication and coupling with circuits. The development of low-dimensional material infrared photodetectors started 40 years ago. In recent

times, most of the reported infrared detectors, such as those based on mercury telluride (HgTe) colloidal quantum dots (CQDs), are still single-pixel devices [5]. Figure 1 shows the history of the development of infrared detectors and room-temperature infrared detectors with low-dimensional materials.

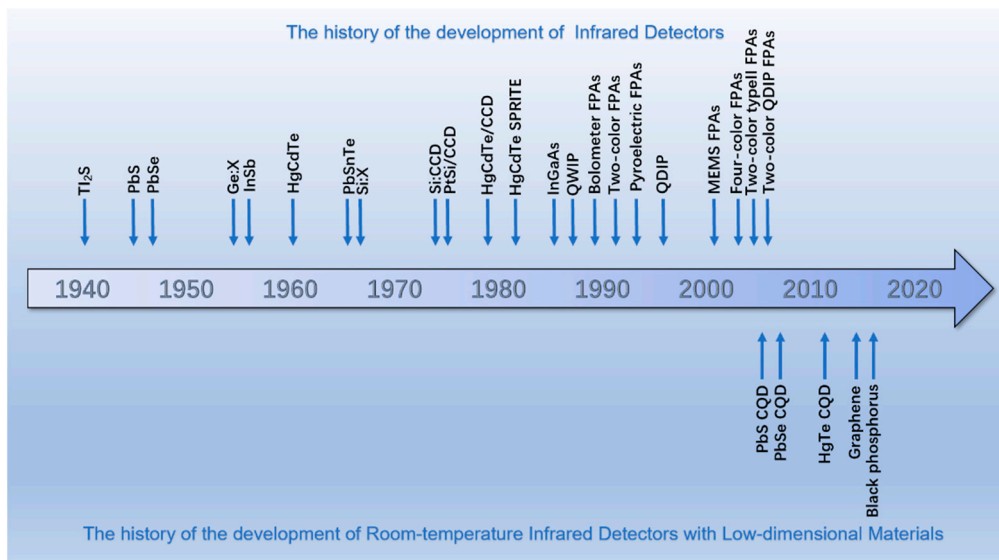

**Figure 1.** "The history of the development of infrared detectors" and "The history of the development of room-temperature infrared detectors with low-dimensional materials".

Low-dimensional materials can be classified into zero-dimensional (0D), one-dimensional (1D) and two-dimensional (2D) materials.

CQDs, which are typically 2–20 nm in diameter, are important representatives of 0D materials [6]. Benefiting from quantum confinement, the band gap of CQDs is strongly dependent on nanocrystal (NC) size, which makes it possible to design a band gap independently. This means that, by controlling the size of CQDs, rather than their chemical composition, a wide spectrum can be covered [7]. For example, the spectra of HgTe CQD can cover short-wave infrared (SWIR), mid-wave infrared (MWIR), long-wave infrared (LWIR) and even Tera Hertz (THz) [8]. Due to the rapid development of material systems, it is now even possible to achieve this wide spectral range using non-toxic and heavy metal-free quantum dots [9]. In addition, the preparation process of CQDs has significant advantages. CQDs are typically made from II-IV, III-V and IV-VI semiconductors through inexpensive and scalable wet chemical synthesis [6]. This enables cheap large-scale manufacturing of optoelectronic equipment at room temperature by several techniques, such as roll-to-roll processing [7]. The flexibility of CQDs is also reflected in the fact that the modern solution of ligand exchange protocols can change the properties of CQDs prior to the deposition process. For example, long-chain organic ligands on the surface of CQDs can be replaced by short-chain organic compounds and even by inorganic ions such as bromides or iodides for better carrier transport [10]. In fact, CQDs have already achieved great success in high-performance infrared detection and in multi-functions such as those in SWIR/MWIR dual band photodetectors [11] and polarization infrared detectors [12] due to unique structure-related photoelectric properties.

In recent years, 1D and 2D nanomaterials have become the focus of nanotechnology research. At present, 1D nanomaterials can be synthesized from a single crystal form, whose chemical composition, shape, doping state, length, diameter and other key parameters are controllable [13]. Now, near-infrared (NIR) photodetectors based on MBE-grown InAs NWs show high-performance optoelectronic properties. Single InAs NW photodetectors have an impressive $I_{on}/I_{off}$ ratio of $10^5$ with a maximum field-effect mobility of $\sim$2000 cm$^2$/V·s [14]. For 2D materials, graphene is one main research direction in infrared detectors.

A waveguide-integrated graphene photodetector that simultaneously exhibits high responsivity, high speed and broad spectral band width can achieve a photoresponsivity exceeding 0.1 AW$^{-1}$ together with a nearly uniform response between 1450 and 1590 nm [15]. Another 2D material of interest is black phosphorus. Black phosphorus MWIR detectors have been demonstrated at 3.39 μm with high internal gain, resulting in an external responsivity of 82 A/W [16]. In addition, there is a lot of research on the physicochemical properties of 2D electron gas [17–19].

Excellent light response and signal-to-noise ratios are indispensable for obtaining an excellent room-temperature detector. For PC devices, 1/f noise and shot noise are inevitable because of bias voltage. For PV devices, 1/f noise and Johnson noise are the main interference signals when there is no bias voltage, whereas shot noise has a greater effect than Johnson noise when bias voltage is applied.

In this review, we excessively introduce room-temperature infrared detectors based on lead sulfide (PbS), lead selenide (PbSe) and HgTe CQDs, which belong to 0D materials, as well as graphene and black phosphorus, which belong to 2D materials. There have been good review articles on room-temperature detectors and materials before. The review of PbSe thin-film infrared detectors reported by Gupta et al. in 2021 [20] introduces the physical and chemical properties of this material in detail and summarizes the latest progress in the research of PbSe photoconductivity. The review reported by Wang et al. in 2019, which introduces mature technology of large-scale commercialization, newly developed technology based on quantum cascade photodetectors, and brand-new concept devices based on 2D materials, provides a comprehensive view of the progress and challenges of room-temperature infrared detectors [21]. Reviews of room-temperature photoelectric infrared detectors focusing only on a certain material have more detailed descriptions of the properties and applications of this material, such as a review reported in 2019 which focuses on a hybrid structure based on 2D materials [22]. A review reported in 2018, which summarizes emerging technologies for high performance IR detectors, takes the material type as the framework, which is similar to our review [23]. This review aims to cover more materials for room-temperature IR photodetectors.

## 2. PbS

The development history of PbS and PbSe is introduced here because the developments of various lead salt materials are closely related to each other. As a result, the development course of PbSe is not repeated in Section 3.

The earliest information about lead-based semiconductor materials comes from a patent published in 1904 by Bose, who found and utilized the photovoltaic effect of a crystal of galena. Subsequently, Case carried out his research on thin films of thallous sulfide (Tl$_2$S) in 1917 [24] and 1920 [25]. Due to the military needs of infrared information in World War II, Germany developed lead salt (PbS, PbSe and lead telluride (PbTe)) materials vigorously in the 1930s. During that period, different methods for preparing lead salt thin films developed rapidly. Gudden and Kutzscher prepared lead salt films by evaporation and chemical deposition, respectively. Shortly after German scientists firstly studied it, the United States scientists also conducted research on it. Cashman of Northwestern University began work on Tl$_2$S in 1941 and later turned his full attention to the preparation of thin films of PbS, PbSe and PbTe by vacuum evaporation. Among the three typical lead salts used in infrared detectors, PbS and PbSe have been developed and produced to some extent, but PbTe has not been adapted for production and has been gradually phased out [26]. The cut-off wavelength of PbS and PbSe QD materials is short wave, and the research on them is still in the laboratory stage. Additionally, it is worth mentioning that mature commercial products based on PbS and PbSe bulk materials with a cut-off wavelength of 3–5 μm have been around for a long time. At present, lead salt detectors on the market are mainly single-point detectors with an mm2 detection area level. Lead salt photodetectors, which are competitive in the market, commonly show the detectivity of 10$^{10}$ Jones.

### 2.1. Physical Properties of PbS

The study on the optical properties of PbS mainly focuses on single or polycrystalline films in the early stage. Many studies have shown differences in light scattering and absorption in nanostructures compared to bulk materials. Therefore, the particle size effect on the optical properties of semiconductors has attracted much attention.

Exciton, formed by direct light excitation or by the combination of free carriers [27], is an important concept in understanding nanoparticle properties. The delocalization region of excitons may be much larger than the semiconductor lattice constant, which may even be several or tens of nanometers. Although nanoparticle sizes are comparable to exciton radii, their physical properties are greatly changed [28]. In the case of PbS nanoparticles, the band gap in the electron spectrum increases from 0.41 to 1.92 eV when the particle diameter is reduced to 5 nm, and the band gap continues increasing when the diameter is further reduced [29]. This indicates that the band gap increases with the decrease in semiconductor particle size, which means a blue shift of the absorption band. There are several reports on optical transmission properties of PbS nanoparticle films, which is summarized in the following section.

#### 2.1.1. A Quantitative Estimation Method for Band Gap Width

To determine the width of the band gap, the most efficient way is to observe the part of the spectrum where the transmittance varies significantly with wavelength. Experimental data show that the most significant dropping part is the region from 700–800 to 2500–2600 nm corresponding to the photon energies from ~1.6 to ~0.5 eV.

Quantitative estimation of semiconductor band gaps by the absorption spectrum [28] depends on the absorption coefficient, which indicates the absorption rate of 1 cm-thick material. The formula is as follows:

$$\sigma = -logT/s \tag{1}$$

where absorption coefficient $\sigma$ is the optical density associated with a 1 cm-thick layer of material, $T$ stands for the transmittance in relative units, and $s$ is the material thickness, with unit cm.

In order to investigate the relationship between the spectral dependence of the absorption coefficient and the semiconductor band gap, the following formula is introduced [30]:

$$(\sigma(\omega)\hbar\omega)^{1/n} = A_n(\hbar\omega - E_g) \tag{2}$$

where $\sigma(\omega)$ is the spectral dependence of the absorption coefficient; $\hbar$ is the reduced Planck constant, which is used to describe quantum size; $\hbar\omega$ is the photon energy; $A_n$ is a coefficient that is dependent on the type of transition and independent of $\omega$; and $E_g$ represents the semiconductor band gap. Then, there is the discussion of the choice of $n$. The value of $n$ depends on the band gap type: $n = 1/2$ for the direct allowed transition; $n = 3/2$ for the direct forbidden transition; $n = 2$ for the indirect allowed transition; and $n = 3$ for the indirect forbidden transition. If it is based on reflectance spectroscopy, $\sigma(\omega)$ should be replaced by $F(R_\infty)$, which is understood by the following formula:

$$F(R_\infty) = \frac{(1-R)^2}{2R} \tag{3}$$

where $R$ is the reflection coefficient.

When $n = 1/2$ and the electron–hole interaction is ignored, Formula (2) can be simplified as follows:

$$(\sigma(\omega)\hbar\omega)^2 = A^2(\hbar\omega - E_g) \tag{4}$$

Formula (4) is more convenient for data processing and band-gap calculation. However, if the electron–hole interaction cannot be ignored, the spectral dependence of the absorption coefficient makes more sense in the following form [31]:

$$\sigma(\omega) = \frac{B}{1 - exp\left[-C\left(\hbar\omega - E_g\right)^{-1/2}\right]} \tag{5}$$

where *B* and *C* are constants. However, in PbS films, due to a high dielectric constant, the electron–hole interaction is not obvious. As a result, Formula (4) is a good choice for the direct allowed transition.

For monodisperse nanoparticles, the absorption spectrum has an easy-to-understand smoothing function with respect to wavelength or energy. However, thin films normally take on the form of polydisperse particles. As a result, Sadovnikov et al. proposed a model to solve this situation [32], which is as follows:

$$\sigma(\omega) = \sum_i c_i \sigma_i(\omega) = \sum_i \frac{c_i A \left(\hbar\omega - E_{g_i}\right)^{1/2}}{\hbar\omega} \tag{6}$$

where $c_i$ is the relative number of nanoparticles of a given size in a film and $E_{g_i}$ is the band gap corresponding to the above nanoparticles. Different grain sizes correspond to different *i* in PbS nanoparticle films. The final absorption spectrum is the superposition of the corresponding spectra of nanoparticles of different sizes (or different *i*). Sadovnikov et al. deduced mathematically that the visible band gap $E_{g_\Sigma}$ is an additive function of the corresponding quantities of nanoparticles of different sizes [32]. By transforming Formula (6), we can obtain

$$[\sigma E]^2 = A^2 \left[\sum_i c_i \left(E - E_{g_i}\right)^{1/2}\right]^2 \tag{7}$$

In addition, due to $[\sigma E]^2 = A^2 \left(E - E_{g_\Sigma}\right)$, we can simplify Formula (7) to

$$E_{g_\Sigma} \approx \sum_i c_i^2 E_{g_i} + 2\sum_{i \neq j} c_i c_j E_{g_i} E_{g_j}{}^{1/2} + \frac{\left|E_{g_i} - E_{g_j}\right|}{3} \tag{8}$$

When the film has monodisperse nanoparticles $c_1 = 1$ and $c_{i \neq 1} = 0$, in this case, the Formula (8) can be reduced to a boundary condition:

$$E_{g_\Sigma} = E_{g_1} \tag{9}$$

This highly developed interface between the grains of continuous thin films makes the positions of conduction bands and valence bands in the electronic structures of the grains fuzzy, resulting in uncertainties regarding the band gap. The effect of this property on the film spectrum is similar to the difference in the size of nanoparticles [32].

### 2.1.2. Properties

The photo-response of photodetectors based on PbS thin films has been fully studied by many groups. Here, we provide data measured by Reisfeld et al., who used current-voltage (I–V) characteristics under different lighting intensities, as shown in Figure 2a,b.

Peterson et al. prepared PbS CQDs according to the method of Hines and Scholes [33,34]. They found that these CQDs exhibit distinct exciton peaks in their absorption spectra and band edge luminescence, which can be tuned throughout the NIR region [33] (Figure 2c). To make the measurements using a Si charge-coupled detector (CCD), they synthesized PbS CQDs with the smallest possible particle size, which fluoresced at wavelengths < 1000 nm. The maximum absorption and emission values of the first exciton are 780 and 930 nm, respectively, with a full width at a half-maximum of 170 meV [33] (Figure 2c). Brown et al.



confirmed that the energy band of the CQDs can be changed through ligand exchange, resulting in energy level drifts of the CQDs as high as 0.9 eV [35]. Figure 2d shows the chemical structure of the ligand they used, and Figure 2e shows the complete energy level diagram of PbS CQDs undergoing ligand exchange. Tang et al. studied the effect of the two-step ligand exchange method on the light response of PbS CQDs [36]. The absorption spectra and emission spectra of CQD films prepared by the classical one-step method and the optimized two-step ligand exchange method are shown in Figure 2f. Ushakova et al. studied the control of the process of CQD self-assembly, superlattice uniformity and interparticle distance by optimizing the number of surface ligands. In addition, annealing and aging also affect the structural stability and optical properties of superlattices [37]. Figure 2g illustrates the evolution of small-angle X-ray scattering (SAXS) patterns of superlattices (SLs) formed by 4.3 nm QDs with the time of sample annealing. The interference peak corresponding to the ordered QD ensemble vanishes, and the optical properties of the SLs undergo changes in parallel. In Figure 2h, the absorption and photoluminescence (PL) spectra of SL QD$_{4.3}$ are compared with those of the initial 4.3 nm QD solution. After 1 min of annealing, the absorption and PL bands become broader, and the PL intensity decreases [37].

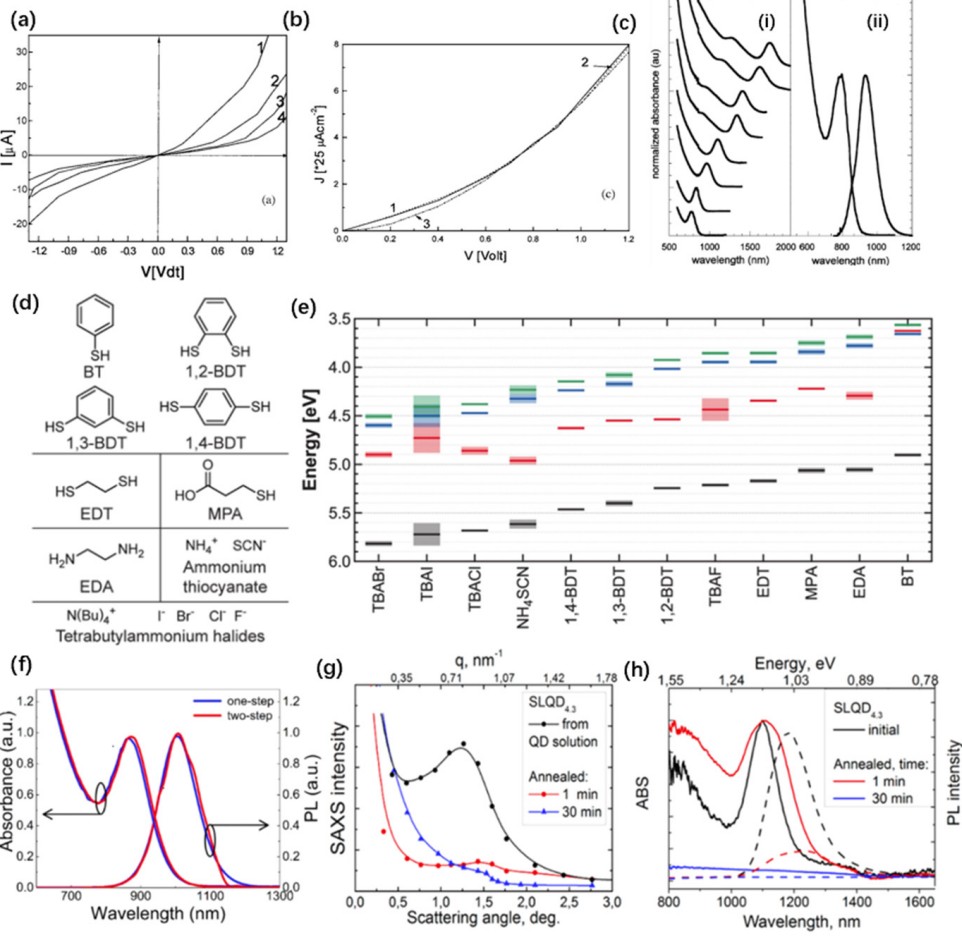

**Figure 2.** (**a**) I-V characteristics of PbS NCs measured with different NCs mole %. Reprinted with permission from Ref. [38]. Copyright 2002 Springer Nature. (**b**) I-V characteristic of PbS NCs measured at various temperatures. Reprinted with permission from Ref. [38]. Copyright 2002 Springer Nature. (**c**) (i) Absorption spectra of PbS CQDs with different sizes. (ii) Typical absorption (left curve) and fluorescence (right curve) spectra of PbS QDs. Reprinted with permission from Ref. [33]. Copyright 2006 American Chemical Society. (**d**) Chemical structures of the ligands studied by Patrick. Reprinted with permission from Ref. [35]. Copyright 2014 American Chemical Society. (**e**) Energy level diagrams

of PbS CQDs after ligand exchange, which is shown in d. Reprinted with permission from Ref. [35]. Copyright 2014 American Chemical Society. (**f**) The absorption and fluorescence spectra of PbS CQD films were studied by one-step (blue) and two-step (red) ligand exchange methods. Reprinted with permission from Ref. [36]. Copyright 2019 American Chemical Society. (**g**) SAXS pattern of SLs formed by 4.3 nm QDs: initial (black), annealed for 1 min (red) and annealed for 30 min (blue). Reprinted with permission from Ref. [37]. Copyright 2016 American Chemical Society. (**h**) Absorption (ABS) (solid curves) and PL (dashed curves) spectra of SLQD$_{4.3}$: initial (black), annealed for 1 min (red) and annealed for 30 min (blue). Reprinted with permission from Ref. [37]. Copyright 2016 American Chemical Society.

### 2.2. Fabrication Methods

There have been many mature methods on the preparation of PbS CQDs. For PbS CQDs, there are two major organic synthesis methods [34].

One way is to react lead oleate and bis(trimethylsilyl) sulfide (TMS) in octadecene (ODE). They produce monodisperse CQDs that are 2.6 to 7.2 nanometers in size with corresponding absorption peaks ranging from 825 to 1750 nanometers. However, not all sizes of CQDs made in this way are air-stable [39]. The following are the detailed operation steps: First, the lead oxide (PbO) in oleic acid (OA) is heated under argon or vacuum to prepare lead oleate. After that, a solution of TMS in ODE is injected into the solution until the ratio of lead to sulfur is 2:1. It is worth mentioning that trioctylphospine (TOP) can also be used as a solvent for TMS and has no material effect on the reaction results. The reaction temperature is controlled according to the desired particle size. The NCs are then precipitated with a polar solvent such as methanol or acetone and are subsequently redispersed into an organic solvent such as chloroform or toluene. Precipitation and redissolution are repeated to ensure the removal of the reaction solvents. Finally, the aqueous NC dispersion is centrifuged to remove any remaining impurities [34].

The second method is to react lead chloride (PbCl$_2$) and elemental sulfur in oleylamine (OAm) under nitrogen. PbS NCs were prepared by thermal injection, similar to the first method. The size range (4.2–6.4 nm) of CQDs prepared by this method is much smaller than that of the first method. In addition, the corresponding absorption peak range is from 1200 to 1600 nanometers. The CQD films prepared by this method show good optical stability [40]. In addition, there are many preparation methods of PbS CQDs, most of which are based on the improvement of the above two methods.

Preparing PbS CQD solids is further mentioned in many articles, and different authors differ in method details. Here, we briefly introduce a method offered by McDonald et al. It is necessary to carry out ligand exchange first, and this method is an improvement upon Hines's method [34]. They precipitated the OA-coated NCs with methanol, dried them and dispersed them in excess octylamine. The solution was heated, and the octylamine-coated NCs were precipitated with N, N-dimethylformamide and then redispersed into chloroform. The NCs were then mixed with 2-methoxy-5-(2′-ethylhexyloxy-pphenylenevinylene) (MEH-PPV). The P-phenylenevinylene (PPV) hole transport layer was rotated onto an indium tin oxide (ITO)-coated glass sheet and annealed in vacuum to allow polymerization. A mixture of MEH-PPV and NCs was dissolved in chloroform and spun onto the PPV layer to create a thin film. Finally, the upper contact was prepared by vacuum evaporation [41].

### 2.3. Devices

PbS NC films have been used in many devices, including infrared LEDs [42], mid- and long-wave infrared detectors [43], upconversion photodetectors [44] and field effect transistors (FET) [45]. Here are some typical concrete examples.

A hybrid graphene PbS QD phototransistor with ultra-high gain has been reported (Figure 3a). Using the strong light absorption of QDs and the high mobility of graphene, the materials were mixed into a system to make it have high sensitivity for light detection. The ultra-high gain of graphene phototransistors was realized for the first time by using

the charge transfer of the two materials [46]. Coupling with graphene has always been a research direction for the improvement of the performance of detectors.

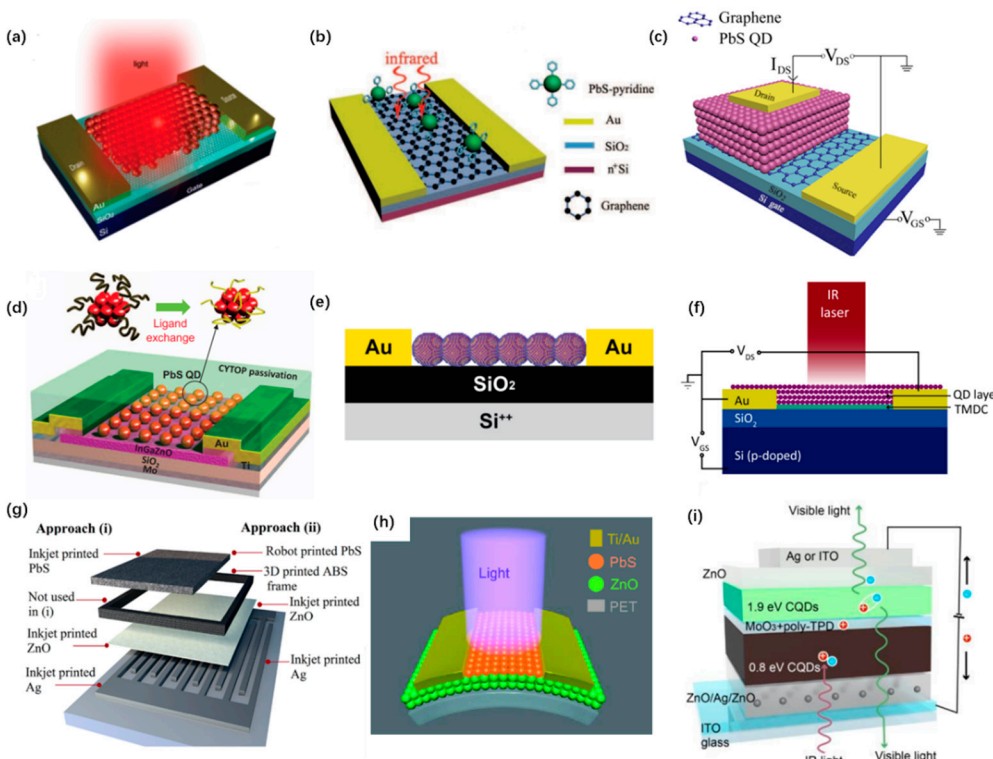

**Figure 3.** (**a**) Schematic diagram of graphene QD hybrid phototransistor; graphene sheets are deposited on the $SiO_2$/Si structure coated with PbS QDs. Reprinted with permission from Ref. [46]. Copyright 2012 Springer Nature. (**b**) Schematic diagram of a graphene transistor modified with PbS QDs under illumination. Reprinted with permission from Ref. [47]. Copyright 2012 John Wiley and Sons. (**c**) The device architecture schematic of the graphene semiconductor metal sandwich construction. Reprinted with permission from Ref. [48]. Copyright 2016 AIP Publishing. (**d**) A schematic 3D view of the ligand exchange process and the PbS QD/IGZO hybrid phototransistor. Reprinted with permission from Ref. [49]. Copyright 2016 Springer Nature. (**e**) PbS CQD FETs based on blade-coated films. Reprinted with permission from Ref. [45]. Copyright 2018 American Chemical Society. (**f**) Hybrid photodetector architecture with circuit diagram. Reprinted with permission from Ref. [50]. Copyright 2019 American Chemical Society. (**g**) Sketch of the photo-conducting devices prepared by two alternative all-printing approaches. Reprinted with permission from Ref. [51]. Copyright 2019 American Chemical Society. (**h**) Schematic diagram of the two-terminal contact flexible photodetector. Reprinted with permission from Ref. [52]. Copyright 2018 Springer Nature. (**i**) Schematic diagram of the structure which uses CQDs as the infrared sensitive layer and CdSe/ZnS core shell QDs as top visible light emission layer. Reprinted with permission from Ref. [44]. Copyright 2020 Springer Nature.

Here is another PbS QD photodetector coupled with CVD-grown graphene (Figure 3b). It shows amazing responsivity of $1 \times 10^7$ AW$^{-1}$ at a power of 30 pW [47]. Graphene was also used in PbS QD field effect phototransistors as electrodes (Figure 3c). PbS QDs are used as channels, and field-effect transistors show good responsivity of $4.2 \times 10^2$ A/W [48]. By comparing specific the detectivity of these PbS–graphene devices, the device reported by Konstantatos et al. shows the best property which reaches a specific detectivity of $7 \times 10^{13}$ Jones.

Besides graphene, many new materials are candidates for coupling materials of PbS devices. A PbS QD-sensitized InGaZnO photoinverter for NIR detection is reported in 2016 (Figure 3d). This hybrid photoelectric device has good light-response performance of

high specific detectivity of $10^{13}$ Jones [49], which is similar to that of the best performing optoelectronic devices based on PbS–graphene, as mentioned before.

In addition, in the research of lead sulfide detectors, there are many novel methods to treat thin films. Single-step fabrication of FET using CQD ink was studied, and the authors of this study also reported the importance of removing ligands after deposition. The structure of the FETs based on blade-coated PbS CQD films is as shown in Figure 3e. They showed that this superficial coating method with blades can prepare high-quality thin films, and the device has a good response [45]. It is also the work of many researchers to make devices by coupling other materials with PbS to expand spectral range. A hybrid PbS CQD transition metal dichalcogenides photodetector with high sensitivity was reported in 2019 (Figure 3f). The authors of this study extended the spectral coverage of this technology from 1.5 μm to 2 μm and showed excellent specific detectivity of $10^{12}$ Jones at room temperature [46].

Perovskite is a hot material that has been used with 3D-printing technology in recent years. As a result, IR photodetectors made of PbS nanotubes with perovskite ligands were studied and reported in 2019 (Figure 3g). This all-printed device shows a cut-off frequency of over 3 kHz and a high detectivity of $10^{12}$ Jones [51], which shows the same order of magnitude as the hybrid PbS CQD transition metal dichalcogenides photodetector, as mentioned before.

The fabrication potential of flexible devices is recognized as a great advantage of CQD materials. A flexible broadband photodetector based on the heterostructure of PbS/ZnO nanoparticles was reported in 2019 (Figure 3h), and it shows the widest detectable spectral range (UV-Vis to NIR) of devices that we introduce in this section [52]. This detector shows a high detectivity of $3.98 \times 10^{12}$ Jones.

Upconversion device is an exciting field because of its unique properties. A QD-based solution-treated upconversion photodetector was studied and reported (Figure 3i). The photodetector has a low dark current, a high detectivity of $6.4 \times 10^{12}$ Jones, a millisecond response time and compatibility with flexible substrates [44].

## 3. PbSe

PbSe thin films are widely used for NIR and MWIR range applications due to their unique physical properties. PbSe polycrystalline thin films are widely used for infrared detectors [53]. Considering that the photosensitivity of thin films is very sensitive to crystallite size, a research group prepared thin films with a thickness of 1.2 μm and different crystallite sizes. After post-processing, the detectivity of the photodetector achieved $2.8 \times 10^{10}$ Jones at room temperature [54]. PbSe has many forms, such as polycrystal, monocrystal and QDs, which have been studied deeply and are widely used in infrared detectors. In order to reduce costs and improve performance, there has been a lot of research on nanotechnology in the past few decades [20]. Photonic applications of CQDs involving lead chalcogenides are mainly associated with PbS and PbSe.

### 3.1. Properties

As lead chalcogenides, PbS and PbSe have many similarities in optical properties. PbSe is a typical direct band-gap semiconductor with a narrow band gap of 0.27 eV at room temperature [55]. The narrow band property of bulk PbSe makes it ideal for MWIR detection. The small electron effective mass in PbSe causes a large Bohr radius of 46 nm, which makes the material ideal for studying quantum size effects observed only in large particles with a small surface-to-volume ratio [56]. Combining these properties makes it possible to precisely alter the band gap and the spectral range of optical photoresponsivity [20].

Using sodium selenite sulfate as the selenium source and lead acetate as the lead source, Begum et al. prepared nanocrystalline PbSe films on glass substrate by the chemical bath deposition (CBD) method. Its optical properties can be referred to in Figure 4a. The results show that the optical absorption of PbSe films increases with increases in deposition temperature. This may be due to an increase in grain size and a decrease in

defects [57]. The $(\alpha h\nu)^2$ vs $(h\nu)$ plots of PbSe thin films are linear over a wide spectral range, as shown in Figure 4b. This indicates that there is a direct optical band gap in the PbSe films [58]. Zhu et al. used pulse the sonoelectrochemical synthesis method to prepare PbSe nanoparticles and estimated the band gap of the materials by optical diffuse reflectance spectroscopy [59]. The mathematical basis they used to estimate the band gap width can be referred to in Section 2.1.1. The band gap of PbSe prepared by this method was 1.10 eV. The normalized $(F(R)\times h\nu)^2$ vs $h\nu$ (eV) of PbSe nanoparticles can be referred to in Figure 4b.

The absorption spectra of PbSe CQDs with different particle sizes were given by Gao et al. (Figure 4d) [60]. They used PbSe CQDs with an average radius of 2.8–3.5 nm and a corresponding band gap of 0.60–0.76 eV, judging from the position of the band edge peak in the absorption spectrum. Thambidurai et al. developed a high-performance infrared photoelectric detector up to 2.8 μm based on PbSe CQDs [61]. They gave the photocurrent and voltage characteristics of photodetectors based on PbSe CQDs with different thicknesses (500, 900 and 1400 nm) in the wavelength range of 1.5–2.8 μm [61]. The photoelectric current and voltage characteristics of PbSe CQDs photodetectors with three different thicknesses and different LED illumination are shown in Figure 4e–h.

Ahmad et al. studied PbSe CQD solar cells with >10% efficiency. To explore energy disorder in both CQD films, Urbach behavior was investigated by tail-state absorption, as shown in Figure 4i. Sharp band tails in a range of 1.0 to 1.25 eV were observed for lead iodide (PbI$_2$)-capped PbSe CQD films deposited via the one-step technique, and they exhibited lower Urbach energy (28 meV). However, Urbach energy in PbI$_2$-treated and tetrabutylammonium iodide (TBAI)-treated PbSe CQD layer-by-layer (LBL) films were 43 and 83 meV, respectively [62].

### 3.2. Fabrication Methods

Photodetectors based on PbSe can be divided into two categories depending on the target wavelength of photosensitivity. In the early days, PbSe intrinsic semiconductors were widely used in MWIR detectors. Bulk PbSe has an optical band gap of 0.27 eV and a sensitive wavelength of 4.4 μm. In recent decades, however, there has been much more discussion about PbSe low-dimensional materials. Scholars have turned their attention to exploiting the quantum confinement effect, referring to the phenomenon that the energy quantization of microscopic particles becomes more obvious with decreases in its space motion limitation size, and the energy level changes from a continuous energy band to a discrete energy level, especially when the ground state energy level moves up and blue shift occurs. A common idea is to reduce the grain size below the Bohr radius. The sensitivity wavelength can be as short as 690 nm, and the band gap can reach 0.18 eV by adjusting the grain size of PbSe [20]. PbSe-based infrared detectors have the potential to span the MWIR, SWIR, NIR and even visible wavelengths.

Massive PbSe semiconductor films prepared by chemical water baths and progress in preparation of PbSe NCs are described here.

### 3.2.1. Chemical Bath Deposition

CBD is a simple and effective method to synthesize high-quality semiconductor thin films without expensive and complex equipment [63]. However, it should be noted that film properties with CBD are greatly affected by precursor fluid composition, bath time, bath temperature and PH value. CBD films are often considered to be deposited on silicon, glass or gallium arsenide (GaAs) substrates with a thickness between 0.2 and 2 μm [20,54,64]. Some scholars believe that a rough substrate surface leads to better deposition [65].

Regarding CBD, there are many different combinations of precursors. In 2003, Hancare et al. prepared PbSe films using lead nitrate and selenosulphate as precursors and tartaric acid as the complexing agent. PbSe thin films were deposited onto cleaned, spectroscopic-grade glass substrates [66]. In 2010, Kassim et al. prepared PbSe films with lead nitrate solution as the lead source, sodium selenate as the selenium source and tartaric acid as the complexing agent [67]. In 2013, Qiu et al. carried out research on the room-temperature PbSe

photodetector, mixing sodium hydroxide, lead acetate and selenosulfate into a precursor solution in a ratio of 12:1:1 [54].

In addition, there are few references about the influence of the timing of the chemical water bath deposition process. Anuar et al. found that the duration of the water bath affects grain size, film thickness and surface roughness. They demonstrated in reliable experiments that grain size, film thickness and surface roughness increase when the deposition time increases from 20 to 150 min. The atomic force microscopy (AFM) images of different samples indicate that the film obtained by 60 min of deposition was uniform and that the substrate surface was covered with good spherical particles [68]. Hone et al. prepared three different samples with deposition times of 30, 45 and 60 min. According to X-ray diffraction analysis (XRD) patterns, they found that the peak intensities and preferred orientations of the crystals were affected by different deposition times. When choosing deposition times of 30 and 60 min, the crystals preferred orientations along the (111) plane, whereas for a deposition time of 40 min, the crystals preferred an orientation along the (200) plane [69].

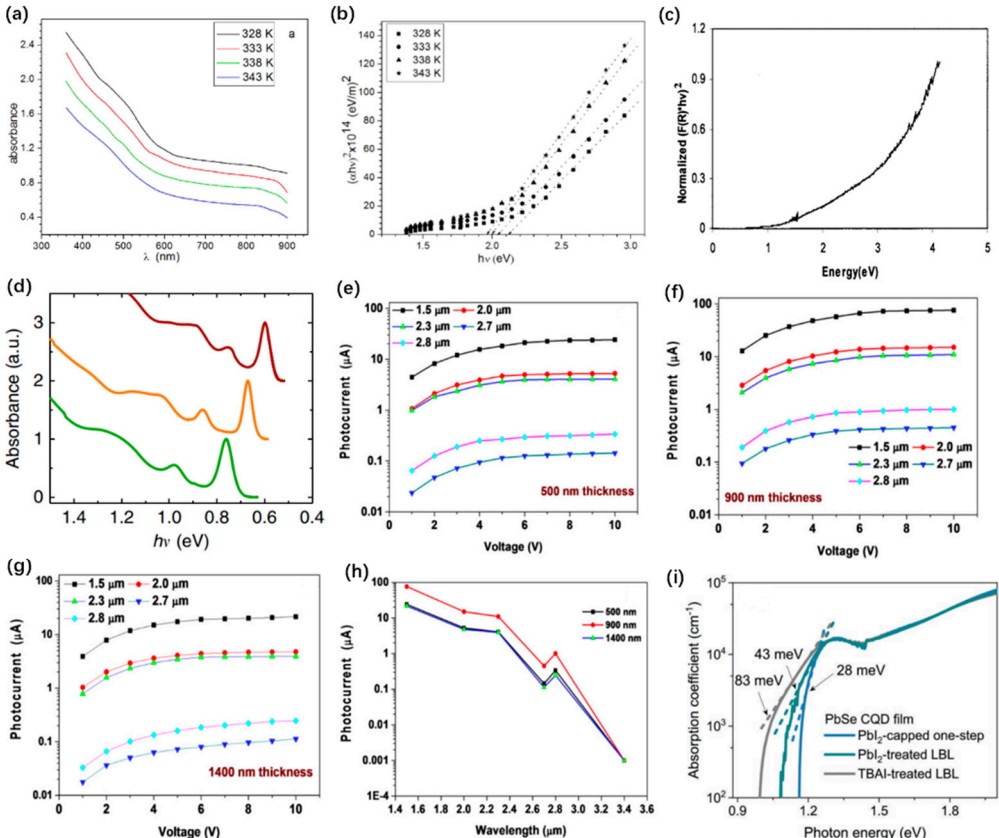

**Figure 4.** (**a**) UV absorption spectra of PbSe thin films reported by Anayara et al. Reprinted with permission from Ref. [57]. Copyright 2012 Beilstein-Institut. (**b**) $(\alpha h\nu)^2$ vs. $(h\nu)$ plots of PbSe reported by Anayara. Reprinted with permission from Ref. [57]. Copyright 2012 Beilstein-Institut. (**c**) Normalized $(F(R) \times h\nu)^2$ vs. $h\nu$ (eV) of PbSe by pulse sonoelectrochemical synthesis method. Reprinted with permission from Ref. [59]. Copyright 2000 American Chemical Society. (**d**) Absorption spectra of three QDs with different radii of 2.8 nm (green), 3.0 nm (orange) and 3.5 nm (red). Reprinted with permission from Ref. [60]. Copyright 2015 Springer Nature. (**e**–**g**) I-V characteristics of devices under different illuminations with wavelengths from 1.5 μm to 2.8 μm for PbSe thin films with different thicknesses. Reprinted with permission from Ref. [61]. Copyright 2017 Optical Society of America. (**h**) The photocurrent of the devices under different illuminations with wavelengths from 1.5 μm to 3.4 μm. Reprinted with permission from Ref. [61]. Copyright 2017 Optical Society of America. (**i**) Urbach energy of three differently treated films. Reprinted with permission from Ref. [62]. Copyright 2019 John Wiley and Sons.

The influence of temperature and PH value on CBD has been widely reported. Deposition temperature has always been considered the most important parameter affecting film quality. It is believed that, with increases in deposition temperature, grain size increases, and dislocation density and microstrain decrease. Reductions in dislocation density and microstrain indicate reductions in lattice defects, i.e., the improvement of film quality. Additionally, deposition temperature has a significant effect on preferred orientation and film thickness [70].

CBD polycrystalline films have good properties when sensitized with oxygen and iodine, which is necessary to activate PbSe as an MWIR detector. However, the specific mechanism of sensitization has not been clearly defined. The sensitization process of PbSe varies from reference to reference, but it usually involves two thermal steps: oxidation and iodization. In order to obtain better performance for PbSe infrared detectors, it is very important to study the mechanism behind the sensitization process, but there are still many doubts.

CBD also shows good prospects in the preparation of PbSe NCs. The chemical characteristics of PbSe thin films are strongly influenced by growth conditions such as ion concentration, PH value and deposition time. Studies have shown that the average size of PbSe nanoparticles increases from 23 nm to 51 nm as the deposition time passes from 1 h to 16 h [71]. In addition, increases in temperature also make the grain size larger [72].

In addition to chemical water bath deposition, PbSe films can also be synthesized by a variety of deposition techniques, such as co-evaporation [73], pulse acoustic electrochemical [59], thermal evaporation [74] and pulse laser deposition [75]. Figure 5 shows the schematic diagrams of CBD and the CQD synthetic procedures.

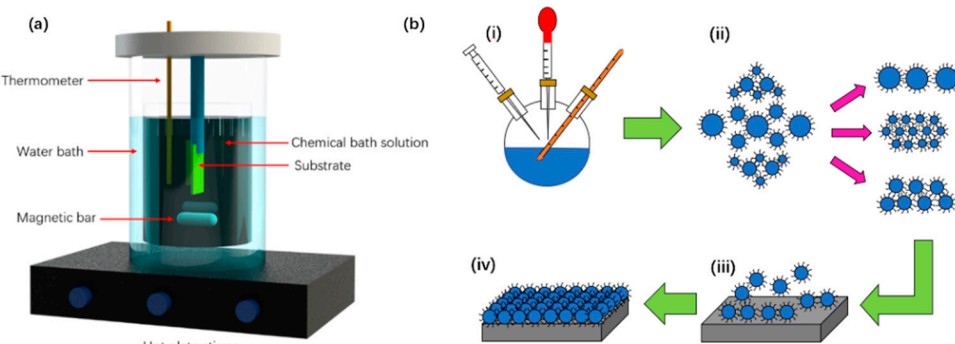

**Figure 5.** (**a**) Schematic diagram of chemical bath deposition. (**b**) Schematic diagram of the CQD synthetic procedures: (i) Synthesize CQDs by solution-phase routes; (ii) Reduce the size distribution of CQD samples by size-selective precipitation; (iii) Deposit CQD dispersions that self-assemble; (iv) Form ordered CQD assemblies.

### 3.2.2. Fabrication of PbSe NCs

Monodispersion is required for photodetectors based on PbSe CQDs. A rapid nucleation followed by a slow growth process is considered the key [76]. Nucleation is affected by temperature, degree of supersaturation in solution, interfacial tension, etc. [77]. There are two ways to stop nucleation. One way is to lower the concentration of the solution below a certain level. The other is to rapidly inject precursors into a high-temperature mixed solution, which has achieved the purpose of rapid cooling, commonly known as thermal injection [78]. Thermal injection is considered to be the most widely used method for synthesizing CQDs.

Murray et al. reported a method using lead oleate as the lead source and trioctylphosphine selenide as the selenium source, and the two are dissolved in trioctylphosphine. The above-room-temperature solution is quickly injected into a fast-stirring solution containing diphenylether at 150 °C. The growth rate of NC can be accelerated by increasing the solution temperature, and NC with the larger size can be prepared at a higher temperature. Solution temperatures of 90–220 °C correspond to NC diameters of 3.5–15 nm. When the

grains reach the target size, the dispersion is cooled, short-chain alcohols are added to flocculate the NCs, and it is then separated from the solution by centrifugation [79].

### 3.3. Devices

Research on PbSe photodetectors is focused on improving efficiency, making large imaging FPAs, manufacturing thermoelectric cooling imaging systems, and making more compact and low-cost systems [20]. There are many types of photodetectors based on PbSe, such as photoconductor [80], phototransistor [81] and photodiode [81].

Regarding the most advanced PbSe photodetector equipment, there are mainly the following kinds: broadband photodetectors using PbSe QD [82], PbSe-based photodiode detectors [83], tandem photodiode detectors [84] and PbSe-based FET detectors [85].

Jiang et al. reported an ultra-sensitive tandem CQD photodetector (Figure 6a,b), which shows maximum detectivites of $8.1 \times 10^{13}$ Jones at 1100 nm and 100 K [83].

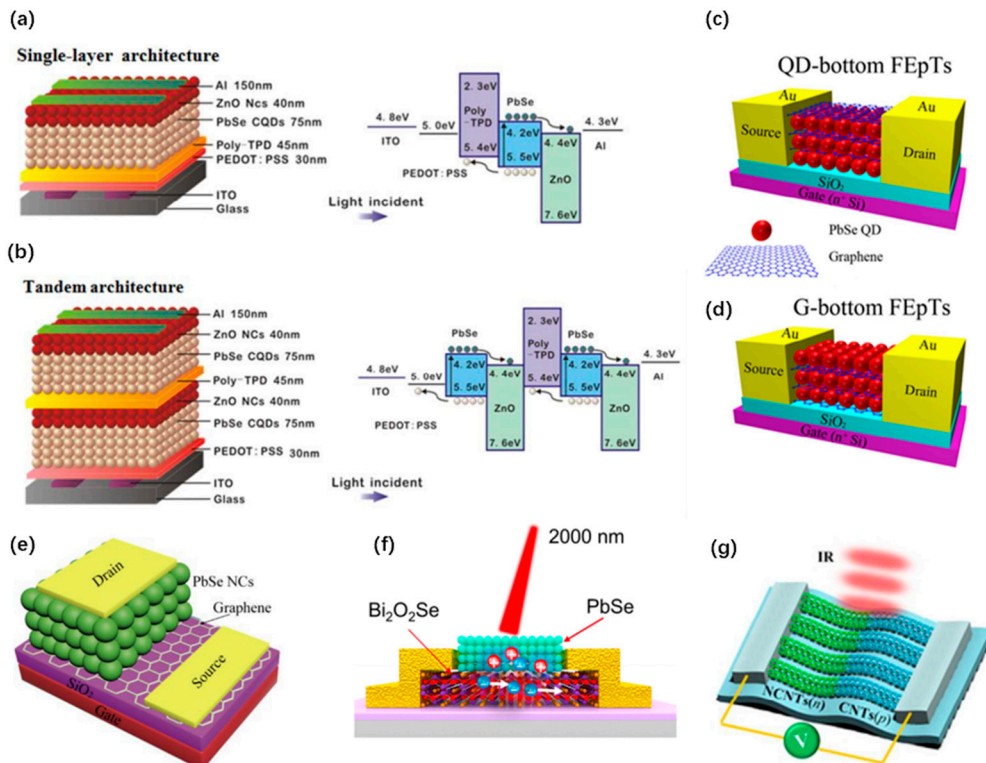

**Figure 6.** (**a**,**b**) Device structures and band diagrams of single-layer and tandem photodetectors. Reprinted with permission from Ref. [83]. Copyright 2015 RSC Pub. (**c**,**d**) Illustrations of QD-bottom FEpTs (**c**) and G-bottom FEpTs (**d**). Reprinted with permission from Ref. [86]. Copyright 2015 American Chemical Society. (**e**) Schematic illustration of PbSe NC-based VPT. Reprinted with permission from Ref. [85]. Copyright 2019 Elsevier. (**f**) Schematic illustration of the PbSe CQD–$Bi_2O_2Se$ nanosheet hybrid photodetector. Reprinted with permission from Ref. [87]. Copyright 2019 American Chemical Society. (**g**) Schematic illustration of the detector's structure. Reprinted with permission from Ref. [80]. Copyright 2016 RSC Pub.

Many research groups have tried to prepare IR photodetectors by coupling graphene with PbSe CQD. A heterojunction phototransistor based on PbSe CQD–graphene hybrids (Figure 6c,d), which shows the highest responsivity of $10^6$ A/W, was reported in 2015 [86]. In addition, the preparation of graphene electrodes on PbSe CQD vertical phototransistors has also been tried (Figure 6e). The phototransistor exhibits an excellent responsivity of $1.1 \times 10^4$ A $W^{-1}$, a detectivity of $1.3 \times 10^{10}$ Jones, and an external quantum efficiency of $1.7 \times 10^6$% [85]. Besides graphene, coupling between other materials and PbSe materials has also been studied and published. A hybrid photodetector based on $Bi_2O_2Se$ nanosheets

sensitized by PbSe CQDs was reported (Figure 6f). Compared to pure $Bi_2O_2Se$ or PbSe CQDs, the interfacial band offset between the two materials enhances the device's responsivity and the response time. This PbSe CQDs–$Bi_2O_2Se$ photodetector can render an infrared response above $10^3$ A/W at 2 μm under external field effects [87].

In addition to coupling a novel two-dimensional material with PbSe QD to obtain a special photodetector, flexible devices are an important development direction of this material. Figure 6g is a schematic diagram of a flexible device which puts n- and p-type carbon nanotubes (CNTs) on a flexible substrate. The IR response of the photodetector is derived from the interface between the n- and p-type CNTs and the Thomson potential. By the CVD method, two kinds of CNT arrays are grown on the substrate and construct the CNT p–n junction [80].

## 4. HgTe

In recent years, HgTe CQDs, as a new choice of infrared detectors, have attracted much attention because of their excellent optical properties, processability and adjustable absorption characteristics. However, at present, reported HgTe CQD infrared detectors mainly focus on short and mid-wave IR photodetectors. Additionally, research progress on LWIR is slow, and there is still a gap between that and commercial detectors. Although the research on this material is still in the laboratory stage at present, its potential in preparation cost, response speed and coupling with flexible substrates is still exciting.

### 4.1. Properties

Keuleyan et al. firstly synthesized MWIR HgTe CQDs [88] and successfully applied them in MWIR photodetection [89]. They prepared the HgTe CQDs by a simple two-step method and found that the absorption, photodetection with sharp edges, and narrow photoluminescence are tunable between 1.3 and 5 μm [88]. Herein, we choose to display the absorption spectra of HgTe CQDs' solutions in $C_2Cl_2$, which shows the relationship between absorption characteristics and sample size (as shown in Figure 7d). Keuleyan et al. found that larger particles give an absorption onset at lower frequencies.

Keuleyan et al. also systematically studied the electronic structure and size-dependent spectrum of HgTe CQDs, and they extended the HgTe CQD spectra covering LWIR [90]. Figure 7a shows the successful extension of the spectrum up to 12 μm at 80 K, which includes the LWIR. Figure 7b shows the absorption spectra of the corresponding films at room temperature.

Allan and Delerue theoretically studied the electronic structure and energy gap of HgTe CQDs [91]. The energy gap of spherical HgTe CQDs is plotted versus size in Figure 7c. It varies according to the form of $1/\left(0.02126 \times d^2 + 0.21562 \times d + 0.01684\right)$ (in electrovolts), where $d$ is the diameter in nanometers [91]. In addition, they came to a conclusion that the gaps of QDs with cubic, tetrahedral or octahedral shapes are almost the same as the gaps of spherical QDs with the same size, according to previous research [92,93].

However, partial aggregations in HgTe CQD always appear in Keuleyan et al.'s methods, causing difficulties in surface modification for those CQDs. This challenge was solved in 2017, when Shen et al. developed a new recipe for nonaggregating HgTe CQDs, which can be stabilized without any thiols [94].

Later, in 2018, Hudson et al. studied the conduction band fine structure of HgTe CQDs. They synthesized highly monodispersed HgTe CQDs and tuned their doping both chemically and electrochemically [95]. Splitting of the intraband peaks was observed corresponding to nondegenerate $1P_e$ states because of the size uniformity of the CQDs and because of strong spin-orbit coupling in HgTe [95]. We borrow the highlighted picture from the article, as shown in Figure 7e.

Later, in 2019, Chen et al. achieved high carrier mobility in HgTe CQD solids with a hybrid ligand exchange method and improved the detectivity in mid-IR photodetectors [96]. Herein, we show detectivity from 80 to 300 K using measured responsivity and noise at each temperature, as shown in Figure 7f. The 120 nm-thick HgTe/hybrid device shows

a higher detectivity than the 260 nm thick HgTe/EDT device at all temperatures. It is indicated that the HgTe/hybrid ligand material is 10 times better than HgTe/EDT at 80 K and 5 better times at 300 K. Since higher mobility allows longer carrier diffusion lengths and higher operation temperatures in the geminate recombination regime, the transport improvement generally benefits CQD photodetection devices [96].

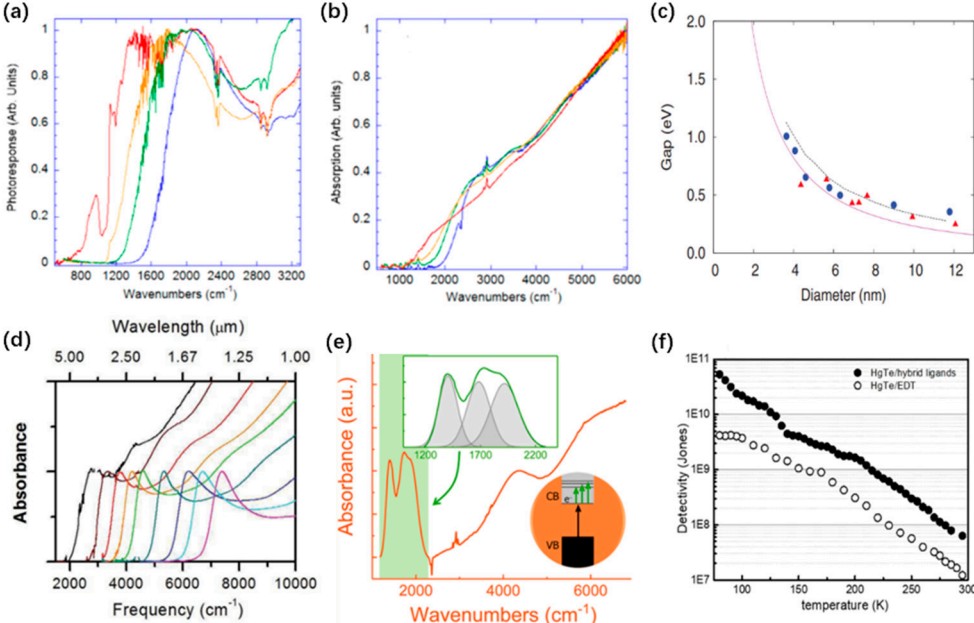

**Figure 7.** (**a**) Photo-response of thin films of HgTe QDs on a Si/SiO$_2$ substrate. Samples were made with different temperatures of 100 °C (blue), 110 °C (green) and 120 °C (orange). The red-line sample was obtained with a regrowth. Reprinted with permission from Ref. [90]. Copyright 2014 American Chemical Society. (**b**) Absorption spectra of thin films of HgTe QDs with the same color scheme as in (**a**). Reprinted with permission from Ref. [90]. Copyright 2014 American Chemical Society. (**c**) Energy gap of spherical HgTe QDs (300 K) versus diameter (magenta solid line) compared to the experimental data of Ref. [97] (red triangles) and Ref. [98] (blue disks). The black-dotted line represents the energy of the second peak in the calculated absorption spectrum. Reprinted with permission from Ref. [91]. Copyright 2012 American Physical Society. (**d**) Absorption of solutions of HgTe CQDs in C$_2$Cl$_4$. Reprinted with permission from Ref. [88]. Copyright 2011 American Chemical Society. (**e**) Spectrum of N-doped HgTe QDs indicates that doping leads to intraband absorbance and restrains the first two excitonic transitions. The green box illustration shows that the total absorption spectrum profile is composed of three small overlapping absorption spectra. The illustration with the orange background shows intra and interband transitions. Reprinted with permission from Ref. [95]. Copyright 2018 American Chemical Society. (**f**) Detectivity as a function of temperature for the HgTe/hybrid and HgTe/EDT films. Reprinted with permission from Ref. [96]. Copyright 2019 American Chemical Society.

### 4.2. Fabrication Methods

HgTe is one of the most difficult nanoparticle compounds to prepare. In order to study the preparation method of II-VI group quantum wells (QWs) and QDs, Brennan et al. focused on the synthesis of the precursors of the target material. They reported the preparation of the solid-state compounds ZnS, ZnSe, CdS, CdSe, CdTe and HgTe from the corresponding M(ER)$_2$ compounds (M = Zn, Cd, Hg; E = S, Se, Te; R = n-butyl, phenyl) and/or phosphine complexes thereof. The idea is to mix the bidentate phosphine 1,2-bis(diethylphosphino)ethane (DEPE) and the M(ER)$_2$ nucleus at a ratio of 1:2 to produce a coordination polymer or a dimeric compound, depending on the material. They found that thermal decomposition of solid compounds produces bulk solid products, whereas

thermal decomposition of liquid compounds produces nanoparticle-sized solid products. The team made HgTe by processing Hg(TeBu)$_2$ through photodecomposition [99].

Rogach et al. studied the preparation of HgTe NCs by wet chemical synthesis [100]. There are two main systems for wet chemical synthesis of cadmium chalcogenide NCs. One system involves coating NCs with trioctyl phosphine (TOP) / trioctyl phosphine oxide (TOPO) in a non-aqueous solution [101]. The other system uses various thiols as a stabilizer in an aqueous solution [102–104]. Rogach's method is to pass hydrogen telluride gas buffered with nitrogen through aqueous N$_2$-saturated mercury(II) perchlorate solutions at a pH of 11.2 with the presence of 1-thioglycerol as an effective size-regulating and stabilizing agent. HgTe CQDs prepared in this way can be precipitated as powders using 2-propanol and then can be capped with thioglycerol on the surface for the purpose of being able to be re-dissolvable in water [100].

Inspired by Huang's work [105], Green et al. reacted mercury(II) bromide and tri-n-octylphosphine telluride to form the intermediate (HgBr$_2$)$_4$·(TeP(C$_8$H$_{17}$)$_3$)$_3$ and eventually the nanosized HgTe, and the intermediates have not been rigorously tested due to their rapid reaction [106].

In 2011, Guyot-Sionnest et al. reported the first synthesis of HgTe CQDs with absorption and emission in the MWIR ranges (>3 μm) [88,89]. They tried to react mercury chloride (HgCl$_2$) dissolved in OAm with telluride (Te) in TOP, as well as mercury(II) acetate with Te dissolved in tri-n-octylphosphine (TOPTe) in alcohol. In 2014, they further extended the absorption edge of HgTe CQDs into LWIR by making a small modification of the growth dynamics with diluted TOP (Te precursor by OAm) [90].

To control CQD growth in an organic solvent, the particles need good surface passivation provided by long chain ligands and slow particle spread to avoid aggregation. In addition, a low temperature is necessary for the synthesis of HgTe CQDs to refrain from growing to bulk sizes. Keuleyan et al. [88] developed methods from Cademartiri et al. [107] for the above two reasons. They formed a viscous mixture with Hg/OAm ratios of about 1:120. Then, they rapidly injected TOPTe to produce Te. Particle size was controlled by the temperature and duration of reaction, with the smallest particles obtained at 60 °C and with the largest obtained at above 100 °C [88].

In 2017, Shen et al. used HgCl$_2$ as the mercury source, bis-(trimethylsilyl) telluride (TMS$_2$Te) as the tellurium source and OAm as the solvent. They used anhydrous tetrachloroethylene (TCE) to quickly cool the reaction. This reaction can be easily scaled up to 1 mmol HgTe by using 2 mmol HgCl$_2$ and 1 mmol TMS$_2$Te. The grain size of the HgTe CQDs can be expanded to 11 nm by adjusting the HgCl$_2$ to TMS$_2$Te molar ratio to 4:1 [95].

*4.3. Devices*

HgTe CQDs IR photodetectors mainly include three categories: photoconductor, phototransistor and photovoltaic devices [5]. Photoconductors have the simplest device structure and can be prepared by the deposition of CQDs on interdigitated electrodes [108]. Compared with photoconductors, photovoltaic devices, which theoretically avoid 1/f noise and dark currents, can improve the sensitivity of the device [5].

Guyot-Sionnest et al. conducted a lot of relative research [88,89,94–97,109–116] and developed the first background-limited infrared photodetection (BLIP) HgTe CQD MWIR photodetector with a specific detection rate of $4.2 \times 10^{10}$ Jones, with a response time in microseconds and with coverage of all infrared spectral regions [111]. In 2018, this team further improved the sensitivity and response speed of photodetectors through new doping methods [117]. Diagrams of these devices are shown in Figure 8.

In 2019, Tang et al. reported the first HgTe CQD flexible SWIR and MWIR dual-band photodetector with a high detectivity D* of $7.5 \times 10^{10}$ Jones at room temperature and a fast response of 260 ns [118]. Integrating the Fabry–Perot cavity with detectors enhances light absorption and the photo-response, with controllable spectral features. (Figure 8d,e). Later, Tang et al. reported a dual-band infrared detector using stacked CQD photodiodes, which provide a bias-switchable spectral response in two distinct bands [11]. It consists of one

SWIR and one MWIR photodiode arranged in an n–p–n structure, with $Bi_2Se_3$ and $Ag_2Te$ as the n and p layers. By controlling the bias polarity and magnitude, it can be rapidly switched between SWIR and MWIR at modulation frequencies of up to 100 kHz with a D* above $10^{10}$ jones at cryogenic temperature.

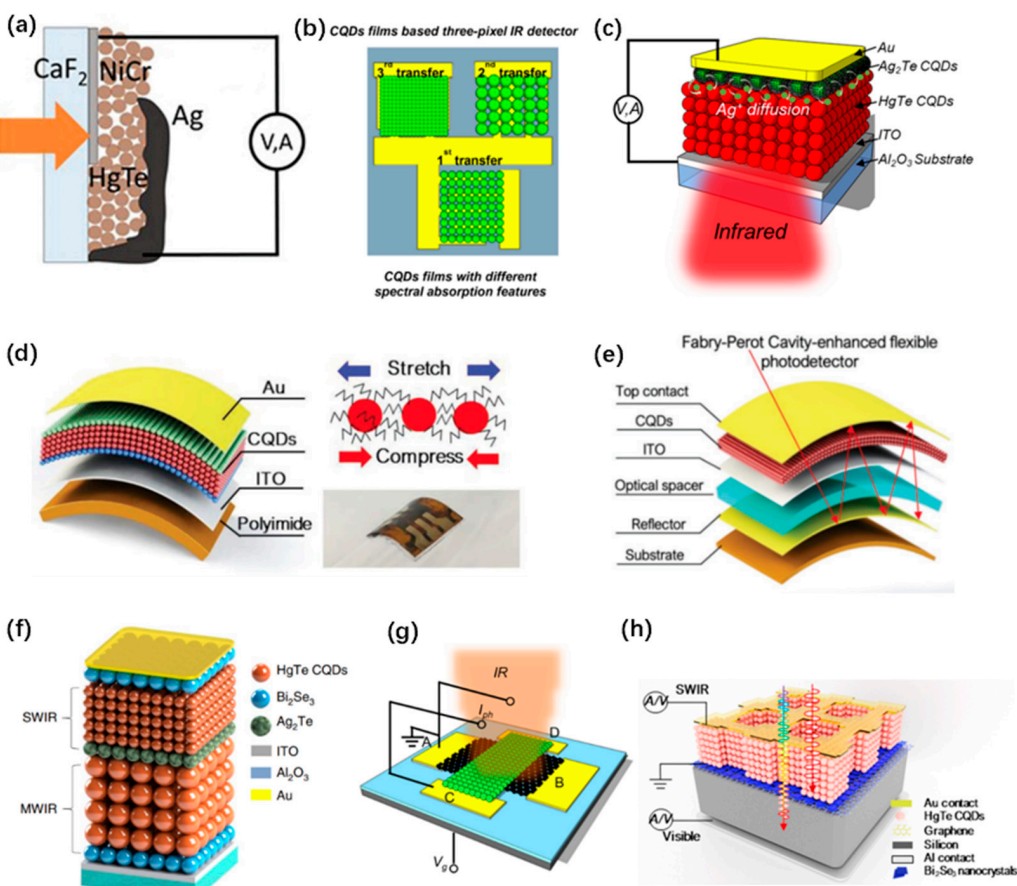

**Figure 8.** (**a**) Schematic diagram of the device structure. Reprinted with permission from Ref. [111]. Copyright 2015 AIP Publishing. (**b**) Schematic configuration of the three-pixel photodetector. Reprinted with permission from Ref. [108]. Copyright 2016 American Chemical Society. (**c**) Photodetector design. Reprinted with permission from Ref. [117]. Copyright 2018 American Chemical Society. (**d**) Schematic diagram of the device architecture of flexible HgTe CQD photovoltaic detectors. Reprinted with permission from Ref. [118]. Copyright 2019 John Wiley and Sons. (**e**) Schematic diagram of Fabry–Perot cavity-enhanced HgTe CQD detectors. Reprinted with permission from Ref. [118]. Copyright 2019 John Wiley and Sons. (**f**) Schematic diagram of the structure of a dual-band CQD imaging device. Reprinted with permission from Ref. [11]. Copyright 2019 Springer Nature. (**g**) Schematic of the graphene/HgTe CQD junction. Reprinted with permission from Ref. [119]. Copyright 2019 American Chemical Society. (**h**) A dual-channel photodetector made by depositing a CQD infrared photodiode onto a graphene/p-silicon Schottky junction. Reprinted with permission from Ref. [120]. Copyright 2020 American Chemical Society.

In addition, Tang et al. reported a graphene/HgTe QD photodetector with gate-tunable IR photo-response [119]. The graphene/HgTe QD junction combines the high carrier mobility of graphene and tunable infrared optical absorption of HgTe CQDs, which offers a promising route for the next generation of infrared optoelectronics.

In 2020, Tang et al. also reported a visible and infrared dual-channel photodetector. They deposited a CQD IR photodiode onto a graphene/p Si Schottky junction and obtained a responsivity of ~0.9 A/W in the visible spectrum, and the infrared CQD photodiode had a detectivity of ~5 × $10^9$ Jones at 2.4 μm [120].

**5. One- and Two-Dimensional Materials**

In this section, we shortly introduce one- and two-dimensional materials used for room-temperature infrared photodetectors. Regarding 1D materials, NWs, especially InAs NWs, are of great concerned due to their high electron mobility [121], easy-to-form ohmic metal contact [122], high electron saturation rate and tiny diameter, which make them mechanically flexible [123]. It has been reported that nanorods are annealed to improve their crystal structures and electrical properties. In addition, detectivity can reach $10^{13}$ Jones, and the noise equivalent power can reach $10^{-14}$ W [124]. Regarding this material, we do not give more details. In Section 5, we give more introductions of 2D materials, such as graphene and black phosphorus. Moreover, 2D materials have attracted much attention because of their flexibility, broadband absorption and high carrier mobility. However, because of its nanometer-order thickness, 2D materials have weak absorption, which limits the detection performance of IR photodetectors based on 2D materials [22].

*5.1. Properties*

5.1.1. Graphene Properties

It is generally believed that a single atomic plane is a 2D crystal, whereas 100 layers are a film of three-dimensional (3D) materials [125]. However, graphene is a special case because its electronic structure evolves rapidly as its number of layers increases, reaching the 3D limit at 10 layers [126]. Only graphene and its bilayers have simple electron spectra, as they are zero-gap semiconductors with one electron and one hole. For layers 3 and above, the spectrum becomes increasingly complex with several carriers present [127], and the conduction bands and the valence bands begin to overlap significantly [126,128].

Peng et al. believe that the size of graphene quantum dots (GQDs) determines its band gap, which ultimately leads to different luminescence conditions when the size of GQDs increases. They obtained three GQDs in different size ranges (1–4 nm, 4–8 nm and 7–11 nm) by controlling the temperature. The resultant GQDs exhibited versatile PL emission color from blue and green to yellow, as the energy gap decreases from 3.90 to 2.89 eV [129].

Since GQDs are fragments derived from graphene flakes, the GQD layer contributes significantly to vertical size, thus altering luminescence properties [130]. Using carbon black as the carbon source, monolayer and multilayer GQDs were prepared simultaneously in nitric acid medium by the top-down method [131].

5.1.2. Black Phosphorus Properties

The band gap of black phosphorus is the most important parameter for determining its light absorption. Band structures characterize the optical properties of 2D materials, especially those that interact with light [132].

Unlike graphene, black phosphorus is formed from a folded honeycomb lattice, which results in strong anisotropic electron mobility and anisotropic optical responses [133]. The layered structure of black phosphorus can be obtained from bulk crystals by mechanical stripping. The most attractive property of black phosphorus is that its layers depend on direct band gaps. The band gap from ~2.0 eV for monolayer, ~1.3 eV for double layers and ~0.8 eV for triple layers lowers down to 0.3 eV for the bulk state [132]. Figure 9c shows the crystal and electronic structure of bulk black phosphorus [134]. Figure 9d,e exhibit the linear absorption and reflection spectra for a few layers of black phosphorus [135,136].

*5.2. Fabrication Methods*

5.2.1. Graphene Preparation

In recent years, due to the great prospect of graphene, there have been many reports on the preparation methods of graphene films.

There are currently seven reliable methods for the preparation of graphene sheets. They are (i) mechanical cleavage of highly oriented pyrolytic graphite (HOPG); (ii) epitaxial growth on an insulator surface; (iii) chemical vapor deposition (CVD) on the surfaces of single crystals of metals; (iv) arc discharge of graphite under suitable situations; (v) use

of intercalated graphite as the starting material; (vi) preparation of appropriate colloidal suspensions in selected solvents; and (vii) the reduction of graphene oxide sheets [137]. The above methods are described in detail in Section 5.2.1.

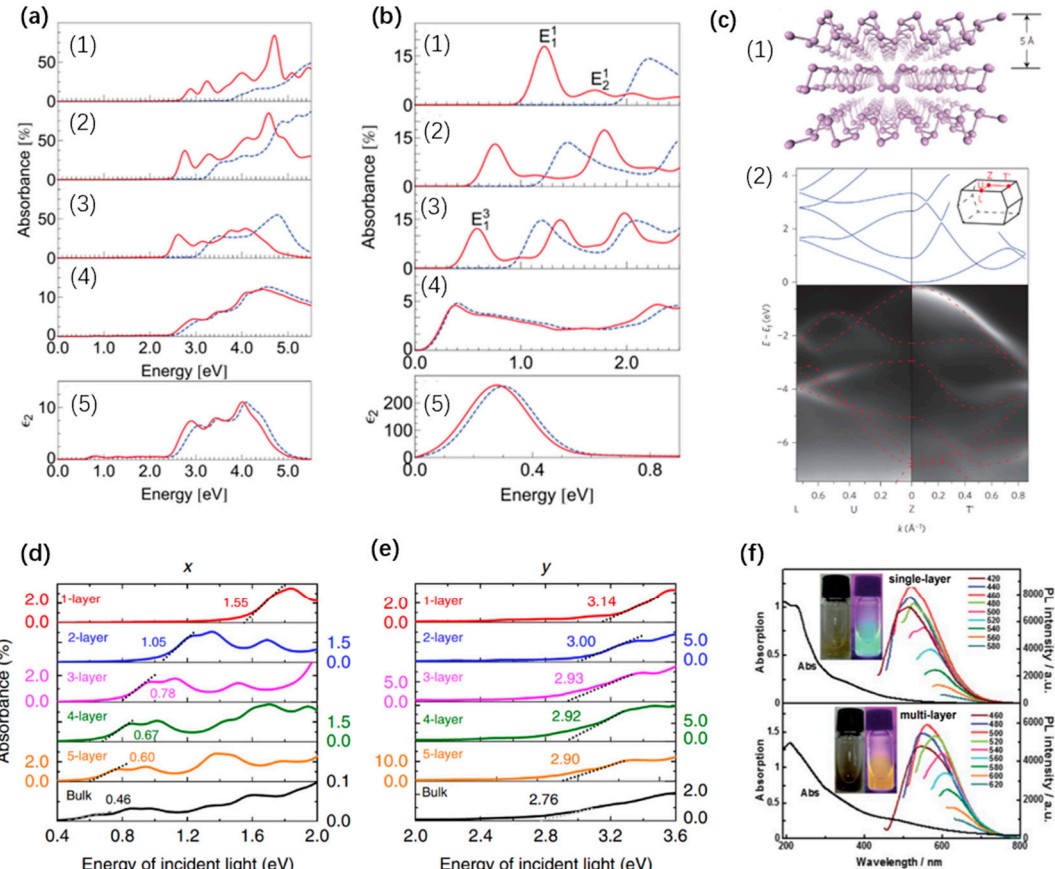

**Figure 9.** (**a**) Absorption spectra of monolayer (1), bilayer (2), trilayer (3), and bulk phosphorene [(4) and (5)] for the incident light polarized along the x (armchair) direction. Reprinted with permission from Ref. [135]. Copyright 2014 American Physical Society. (**b**) Optical absorption spectra of monolayer (1), bilayer (2), trilayer (3), and bulk phosphorene [(4) and (5)] for the incident light polarized along the x (armchair) direction. Reprinted with permission from Ref. [135]. Copyright 2014 American Physical Society. (**c**) Crystal and electronic structure of bulk black phosphorus. (1) Atomic structure of black phosphorus. (2) Band structure of bulk black phosphorus. Reprinted with permission from Ref. [134]. Copyright 2014 Springer Nature. (**d**,**e**) Optical absorption spectra of few-layer BP for light incidents in the c (z) direction and polarized along the a (x) and b (y) directions. Reprinted with permission from Ref. [136]. Copyright 2014 Springer Nature. (**f**) UV-vis absorption and PL emission spectra of GQDs1 and GQDs2 in water solution. Reprinted with permission from Ref. [131]. Copyright 2012 ROYAL SOCIETY OF CHEMISTRY.

For the preparation of mono or by-layer graphene, one of the most common methods is mechanical exfoliation of small mesas of HOPG. Firstly, one side of the HOPG sample is cleaved using scotch tape technology to obtain a clean surface. Then, the other side of the sample is glued to the copper electrode with silver epoxy. As shown in Figure 10b, this copper electrode is connected to the positive terminal of the power supply. The other copper electrode is connected to the ground terminal of the power supply. The second copper electrode is placed with a mica sheet (0.1 mm thick) and a substrate (from top to bottom, a 300 nm-thick $SiO_2$ layer, s 500 μm-thick silicon substrate and a 1 mm glass microscope slide) in turn. When given different voltages, graphene sheets of corresponding thicknesses are cleaved onto the $SiO_2$ layer from the HOPG [138].

Regarding the insulator surface of the epitaxial growth of graphene, SiC is a common choice. High-temperature sublimation of few atomic layers of Si from a mono crystalline SiC substrate is considered the best way to fabricate Few Layer Graphene (FLG) on a full wafer for industrial purposes at present [139]. Epitaxial growth on a substrate is highly dependent on the surface quality of the substrate, because very small defects can affect the growth of the film and reduce its quality. However, commercial SiC substrates have a large number of deep scratches caused by polishing damage. As a result, in order to obtain a good epitaxial growth layer, pretreatment of substrates to eliminate these scratches is particularly important. Hydrogen etching proves to be a good way to eliminate scratches [140]. After $H_2$ etching, samples are heated by electron bombardment in an ultra-high vacuum in order to remove the oxide. After verifying that the oxide is removed, samples are heated to temperatures ranging from 1250 °C to 1450 °C for 1–20 min. Under these conditions, thin graphite layers are formed, with the layer thickness determined predominantly by the temperature [141].

The third method has many similarities with the second one, and the major difference lies in the choice of substrate. According to Fogarassy's idea, graphene is deposited by chemical vapor deposition (CVD) on a Ni (111) thin layer substrate, which is prepared by sputtering Ni on a single crystal sapphire (0001) in an ultra-high vacuum (UHV) [142]. Since a sapphire's surface is difficult to completely clean, grains may be found with 30° rotation in the nickel thin layers. The nickel substrate is then annealed in a hydrogen atmosphere. Finally, the CVD experiments are performed in an atmosphere of a mixture of methane, argon and hydrogen [143].

In 2009, Subrahmanyam et al. introduced a new method to prepare graphene, which can produce 2–4 layers in a relatively large area. Subrahmanyam et al. made a variety of attempts with gas proportions, electrode sizes, discharge currents, etc., in the arc chamber. Only one typical experimental design is introduced here. A graphite rod (Alfa Aesar with 99.999% purity, 6 mm in diameter and 50 mm long) used as the anode and another graphite rod (13 mm in diameter and 60 mm in length) used as the cathode are both placed in a water-cooled stainless-steel chamber filled with a mixture of hydrogen and helium, and the direct current arc discharge of graphite evaporation is to be carried out in this chamber [144]. The discharge current is within the 100–150 A range, with a maximum open circuit voltage of 60 V [145]. The arc is maintained by continuously shifting the cathode at a constant distance from the anode. At the end of the experiment, the deposits collected on the walls of the arc chamber contain only graphene flakes, whereas the cathode deposits contain other impurities [144].

### 5.2.2. Black Phosphorus Preparation

Many methods of preparation of black phosphorus have been reported so far. The synthesis method of black phosphorus can be traced back to 1914. Bridgman et al. placed white phosphorus in a high-pressure cylinder under kerosene. The cylinder pressure was increased from 0.6 GPa to 1.2 GPa, and the temperature was raised from room temperature to 200 °C. After waiting 5 to 30 min, cooling the cylinder and releasing the pressure, the converted black phosphorus could be obtained [146].

Furthermore, Maruyama et al. prepared black phosphorus single crystal in liquid bismuth using white phosphorus as a raw material. The melted bismuth was poured over the white phosphorus and shaken, and then it was maintained at 400 °C for 20 h and was finally slowly brought to room temperature. The solid bismuth was dissolved with nitric acid, and acicular single crystal black phosphorus was obtained in the remaining solution [147].

Shirotani et al. carried out experiments with a wedge-type cubic anvil high-pressure apparatus developed by Wakatuki. They successfully prepared a large black phosphorus single crystal from red phosphorus under a high temperature and high pressure [148].

High pressure was considered a necessary condition for the preparation of black phosphorus, until recently, when reliable references of low-pressure preparation of black

phosphorus were reported. Lange et al. reported a low-pressure method for preparing black phosphorus. They took red phosphorus as a raw material and added a small amount of Au, Sn and a catalytic amount of SnI$_4$ [149]. Nilges et al. gave a more detailed explanation of the method in his article [150]. Köpf et al. further simplified the experiment based on red phosphorus and described the detailed experimental steps in his article [151]. Refer to Figure 10 for the schematic diagram for the preparation of graphene and black phosphorus.

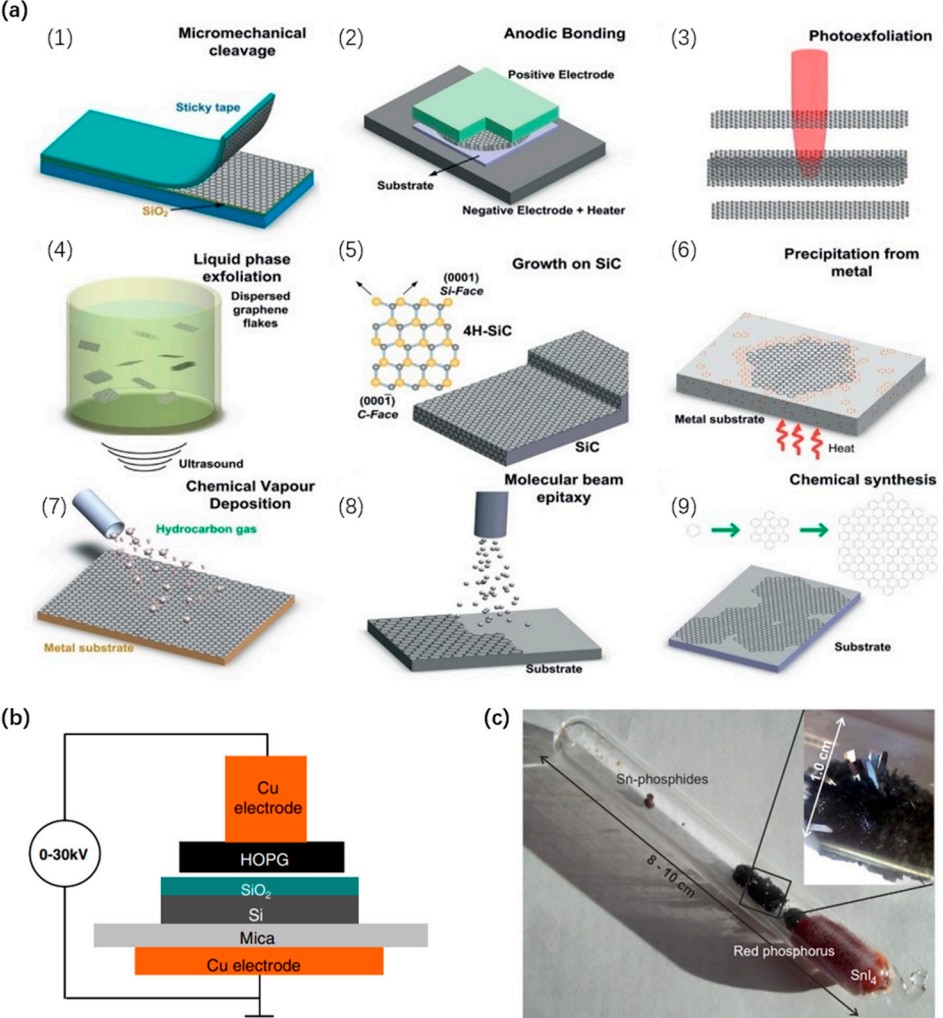

**Figure 10.** (**a**) Schematic illustration of the main graphene production techniques. (1) Micromechanical cleavage. (2) Anodic bonding. (3) Photoexfoliation. (4) Liquid phase exfoliation. (5) Growth on SiC. Gold and grey spheres represent Si and C atoms, respectively. At elevated T, Si atoms evaporate (arrows), leaving a carbon-rich surface that forms graphene sheets. (6) Segregation/precipitation from carbon-containing metal substrate. (7) Chemical vapor deposition. (8) Molecular beam epitaxy. (9) Chemical synthesis using benzene as a building block. Reprinted with permission from Ref. [152]. Copyright 2012 Elsevier. (**b**) Schematic of the experimental setup for electrostatic deposition of graphene sheets. Reprinted with permission from Ref. [138]. Copyright 2007 IOP Publishing. (**c**) A representative silica glass ampoule after the synthesis of black phosphorus. Reprinted with permission from Ref. [151]. Copyright 2014 Elsevier.

### 5.3. Devices

As the hottest material in recent years, graphene has quite a number of applications, such as touchscreen displays [153] (Figure 11f), flexible electronic devices [154] (Figure 11c), organic light-emitting diodes (OLEDs) [155] (Figure 11a,b), high-frequency transistors [156], optical modulators [157], solar cells [158] and ultra-wideband photodetectors [159].

Because monolayer graphene is a zero-band gap material, it can generate photocarriers across a wide electromagnetic spectrum. In the past few years, various graphene-based photodetectors have been reported, such as metal–graphene–metal (MGM) photodetectors, graphene p–n junction photodetectors, graphene–semiconductor heterojunction photodetectors and hybrid photodetectors [159].

As another representative 2D material, black phosphorus has great potential in FET, photoelectric detectors, battery anode materials and thermoelectric applications due to its unique structure and rare anisotropy [160]. In 2014, Liu et al. made the first report on FET based on black-phosphorus-layered materials [161]. It has become widespread research due to its ambipolar FET [162]. Drain-current modulation at room temperature is four orders of magnitude higher than graphene [163] (Figure 11e,g). In addition, black phosphorus has the potential to be used in flexible and wearable products due to its binary properties [164]. Zhu et al. reported a flexible bipolar transistor circuit and an AM demodulator based on black phosphorus [165] (Figure 11d). The low field hole mobility was 310 cm$^2$ V$^{-1}$s$^{-1}$, which is much higher than the most advanced metal oxide flexible transistors.

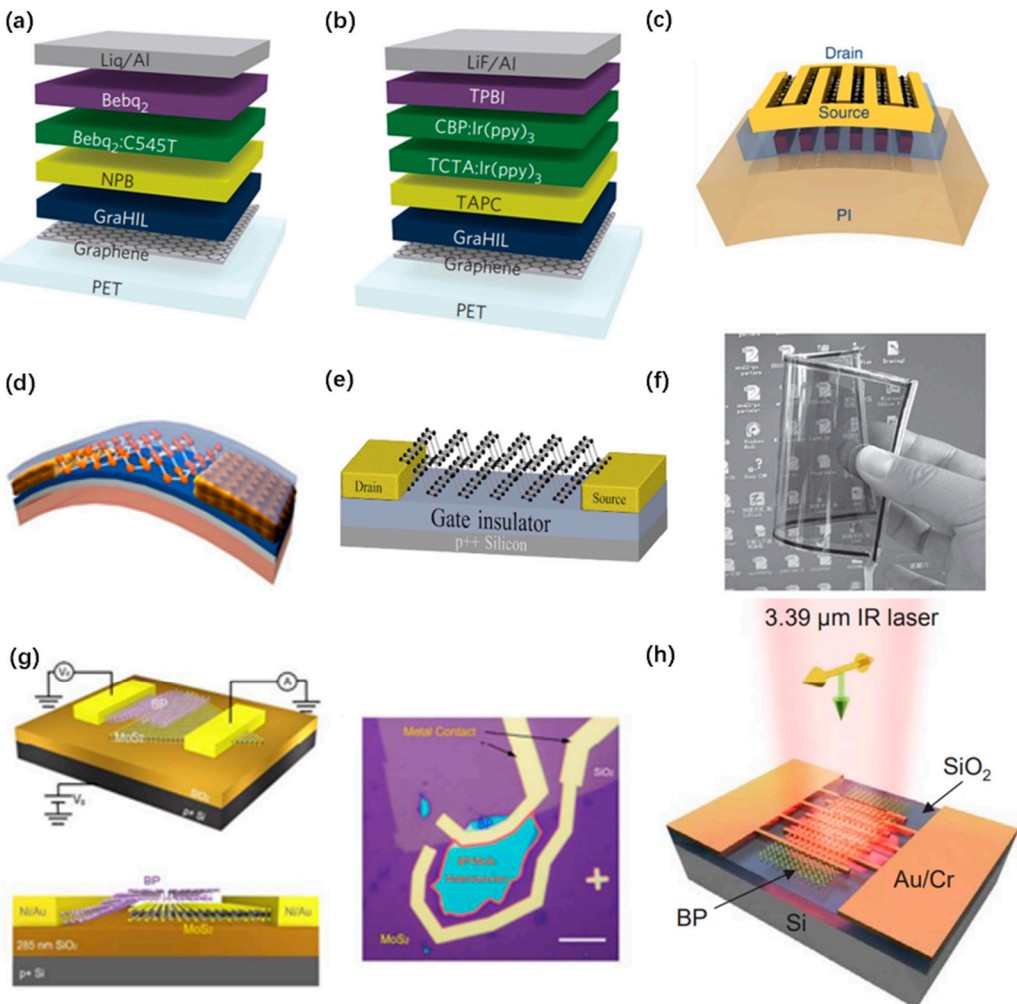

**Figure 11.** (**a**,**b**) Device structures of flexible small-molecule fluorescent OLEDs (**a**) and phosphorescent OLEDs (**b**) using a graphene anode modified with a GraHIL. Reprinted with permission from Ref. [155]. Copyright 2012 Springer Nature. (**c**) Illustration of multifinger embedded-gate graphene device structure that can afford high current drive, thin gate dielectrics, low gate resistance, surface passivation and simple post-transfer fabrication. Reprinted with permission from Ref. [154]. Copyright 2014 Springer Nature. (**d**) Simplified illustration of the device structure of flexible BG BP FET on PI substrate (not to scale). Reprinted with permission from Ref. [165]. Copyright 2015 American



Chemical Society. (**e**) Schematic diagram of the back-gated BP FET. Reprinted with permission from Ref. [163]. Copyright 2015 Royal Society of Chemistry. (**f**) A flexible four-wire resistance touch screen fabricated using RGO/PET as electrodes. Reprinted with permission from Ref. [153]. Copyright 2012 John Wiley and Sons. (**g**) Schematics and optical image of the device structure. Reprinted with permission from Ref. [163]. Copyright 2015 Royal Society of Chemistry. (**h**) Schematic diagram of the BP MSM photodetector operating at 3.39 μm. Reprinted with permission from Ref. [16]. Copyright 2016 American Chemical Society.

It is worth mentioning that mixed-dimensional materials can be a solution in this field. A photodetector based on graphene/$PdSe_2$/germanium heterojunctions has been recently reported. Mixed-dimensional heterojunctions provide enhanced light absorption, and graphene electrodes provide effective carrier collection. Owing to above advantages, this photodetector exhibits great device performance, such as a high specific detectivity and broadband photosensitivity from ultraviolet to MWIR.

In order to better reflect the properties of photodetectors based on different materials, Table 1 shows the performance comparison between the most representative devices in the above mentioned.

**Table 1.** Performance comparison of several representative devices.

| Device | R(A/W) | D*(Jones) | Response Time | Reference |
|---|---|---|---|---|
| Graphene–PbS QD phototransistors | ~$10^8$ | $7 \times 10^3$ | NA | [46] |
| PbS CQD–TMDC photodetectors | $10^7$ | $1.02 \times 10^{12}$ | NA | [50] |
| Tandem PbSe CQD photodetectors | ~0.5 | $8.1 \times 10^{13}$ | NA | [83] |
| PbSe CQD–Bi2O2Se nanosheets hybrid photodetector | >$10^3$ | NA | ~4 ms | [87] |
| HgTe CQD photodetectors | 1.3 | $3.3 \times 10^{11}$ | 50 ns | [117] |
| Stacked CQD photodiodes | NA | $6 \times 10^{10}$ | <2.5 μs | [11] |
| Three-pixel photodetectors | 0.1 | $2 \times 10^7$ | ~ 91.5 ms and ~541.3 ms | [108] |
| HgTe CQD/Graphene/Silicon devices | ~0.9 | ~$5 \times 10^9$ | ~13 ns and ~3 μs | [120] |

## 6. Conclusions

In this paper, the development of room-temperature infrared detectors based on low-dimensional materials is reviewed. We mainly introduce the development of PbS CQDs, PbSe CQDs, HgTe CQDs and some 2D materials. The properties, preparations and device developments of each material are described in detail.

Room-temperature infrared photodetectors theoretically solve a series of problems, such as large volume, high cost and inconvenient use of refrigeration detectors. In addition, the liquid phase preparation of low-dimensional materials also allows low-dimensional materials to be integrated with traditional silicon electronics and flexible large-area substrates at a low cost and with high safety. Additionally, there are still many challenges for IR photodetectors in this field. For example, it is still difficult to achieve a consistent control effect on material synthesis, so the development of precursor chemistry is very important. In addition, learning how to improve multiple-exciton generation (MEG)-enhanced quantum efficiency is also a challenge of great concern. Although there are still many problems to be solved for material growth modes and circuit coupling modes, low-dimensional materials still have broad application prospects, such as subwavelength pixels, large arrays and multicolor devices. Room-temperature infrared photodetectors with low-dimensional materials have already shown great potential in commercialization.

**Author Contributions:** Conceptualization, X.T. and M.C.; investigation, T.L.; writing—original draft preparation, T.L.; writing—review and editing, M.C. and X.T.; project administration, M.C.; funding acquisition, X.T. All authors have read and agreed to the published version of the manuscript.

**Funding:** This research was funded by the National Natural Science Foundation of China and the National Key R&D Program of China (NSFC No. 62035004, 2021YFA0717600 and NSFC No.62105022).

**Institutional Review Board Statement:** Not applicable.

**Informed Consent Statement:** Not applicable.

**Data Availability Statement:** Not applicable.

**Conflicts of Interest:** The authors declare no conflict of interest.

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
