# Peer review of "Room-Temperature Infrared Photodetectors with Zero-Dimensional and New Two-Dimensional Materials"

_coatings, doi:10.3390/coatings12050609_

Round 1
Reviewer 1 Report
The paper aims at a review of a certain class of room-temperature infrared detectors. The topic is important for practical applications.
My comments:
1. The paper does not place the presented work in the proper context. Authors should list the practical applications with requirements of wavelength, detectivity, detector area, acceptable cost etc. and put the subject of the review on this "map".
2. Detectors based on an important class of low-dimensional materials: semiconductor quantum-wells/superlattices are not even mentioned.
It is OK that the authors narrow the scope of an otherwise very broad area but the title and the abstract should reflect the choice and the subject be put in an appropriate context.
3. Important references, key reviews are missing from the references, e.g.:
David Z.Ting et al.
Advances in III-V semiconductor infrared absorbers and detectors
Infrared Physics & Technology 97 (2019) 210-216
Chee Leong Tan et al.
Emerging technologies for high performance infrared detectors
Nanophotonics (2018) 7(1), 169–197
Tiande Liu et al.
Room-temperature infrared photodetectors with hybrid structure based on two-dimensional materials
Chin. Phys. B 28 (2019) 017302
T. Nakotte et al.
Colloidal quantum dot based infrared detectors: extending to the mid-infrared and moving from the lab to the field
J. Mater. Chem. C (2022) 10, 790
A. Rogalski et al.
Comparison of performance limits of the HOT HgCdTe photodiodes with colloidal quantum dot infrared detectors
BULL. POLISH ACADEMY OF SCIENCES - TECHNICAL SCIENCES 68(4) (2020) 845
Wang et al.
Sensing Infrared Photons at Room Temperature: From Bulk Materials to Atomic Layers
Small (2019) 15, 1904396
Authors should carefully study these reviews, and explain clearly how the present paper manifests a different focus, more detailed discussion or more up-to-date view.
The majority of the referenced papers are from the 2014-2019 period, the coverage is small for the most recent advances.
4. I miss a kind of lucid comprehensive summary at the end in the form a table or diagram where the reviewed materials/structures are depicted as a function of sensitivity wavelength, advantages/disadvantages, technological status (laboratory/commercial), existing or potential applications etc.
5. The English of the paper needs improvement. The way the authors abbreviate the journal names in the reference list is not acceptable. Line 183: chapter number should be 2.1.2.
Reviewer 2 Report
Dear authors,
The applications of narrow gap semiconductors as infrared photodetectors are presented. The preparation of PbS, PbSe and HgTe and the device fabrication complemented with its performance is partially presented. Although the scope of the review article is interesting, the critical viewpoints on the said topic are lacking. The listed comments may be considered to improve the visibility of the review article.
Major comments
(1) The figure captions should be simplified. The discussions related to the figures may be improved for more understanding.
(2) The pros and cons associated with the preparation method and the device efficiency should be discussed in detail.
Minor comments
Abstract
(1) The array of nanomaterials spotlighted in this review article may be indicated for clarity.
(2) Some information related to the diverse aspects dealt in the review article may be highlighted.
Introduction
(1) The previous review articles may be briefly discussed followed by bridging the gap analysis and scope of the present review article.
Section 2.0
(1) Line 94: Avoid the term ‘module 3’ and reframe the phrase accordingly.
(2) The size of Tl2S may be provided (if available in the report).
Sections 2.0-4.0
(1) Some recent reports related to the quantization of PbS band gap may be presented for updating the section. Also, corresponding preparation methods may be briefly presented.
(2) Some information on correlating the thickness of PbS/PbSe films, band gap and photodetection performance should be included.
(3) The literature survey on the preparation of PbS using TOPO and OAm assisted methods should be expanded to offer more information on the preparation method. Also, stability as well as photoluminescence studies should be included.
(4) Line 264: The ‘Hines’s method’ may be supported with suitable literature.
(5) Section 2.3: The mentioning of research group identity may be avoided and sentences may be reframed to offer good readability. The similarities and differences among the cited reports may be presented to offer deeper insights into the respective works.
(6) It would be meaningful if chemical reactions associated with CBD towards the product formation and the influence of various reaction parameters over the size distribution of quantum dots is also explained.
(7) Lines 449-451: More information should be provided on the preferred orientations of crystals.
(8) Line 525: sped?
(9) Some general information on the HgTe may be provided.
(10) The gap analysis and the future perspectives in this field may be provided in a separate section.

Reviewer 3 Report
Decision:
Major revision
Comments
The authors reported the Room-temperature Infrared Photodetectors with Low-dimensional Materials, this review can be useful for the scientific community after some revision is highly required. The motivation and goal of the review article are not concisely presented
The authors should address the following points outlined below to improve the scientific quality and provide more data. After the suggested revisions are carefully addressed, this work may be considered for publication.
- The title is not clear as it focuses only on the low-dimensional materials. However, the review has targeted a broad range of materials
- In the abstract section, the author should clearly mention the main points of this review article.
- Most of the Articles concentrate on the basic properties of 0D,1D, and 2D material. They should focus on the basic fabrication process, device structure, and different materials, used for Photodetectors.
- In the material section, the author can compare photodetectors with different materials.
- They should focus on Infrared Photodetectors rather than different properties of materials.
- The conclusion is not clear enough.
- The author should cite some new articles, http://dx.doi.org/10.1007/s13204-021-01787-7
- The author needs to recheck the manuscript for grammatical errors.

Reviewer 4 Report
This review manuscript of “Room temperature infrared photodetectors with low-dimensional materials” summarized the recent advances in the preparation methods and characterization of several materials such as PbS, PbSe, graphene, black phosphorous, etc… It is quite clear about the state-of-the art regarding the experiment realizations based on device fabrication process.
However, it is lack of an important discussion such as the ideal signal-to-noise ratio, and photo-response as mentioned at the beginning. This should be a first step to describe the need of room-temperature photodetector.
Together, the author also needs to have a briefly discuss about the challenges and perspective for future IR photodetector.
Another point should be mentioned is potential of mix-dimensional materials, which could be a solution in this field.
Therefore, I recommend the author to revise these issues before considering for publication.
Round 2
Reviewer 1 Report
The authors had properly addressed the issues raised in the review. However, amending the title (similarly to the way the Abstract was changed) would distinguish the paper from the similar reviews. Also, I still miss an easily comprehensible, graphical or tabular summary of the results.
The English of the paper has also been improved. There are still, however, several minor misprints which must be corrected.
Reviewer 2 Report
Dear authors,
Although few comments are ignored, the revised version still finds interesting and improved corrections offer adequate information on the said topic.
Author Response
Thank you very much for your comments.
Reviewer 3 Report
The authors revised the manuscript very well and hence I agree for publishing this article in its current form.
Author Response
Thank you very much for your comments.